# Sequence Characteristics and Expression Analysis of the Gene Encoding Sedoheptulose-1,7-Bisphosphatase, an Important Calvin Cycle Enzyme in Upland Cotton (*Gossypium hirsutum* L.)

**DOI:** 10.3390/ijms24076648

**Published:** 2023-04-02

**Authors:** Maoni Chao, Genhai Hu, Jie Dong, Yu Chen, Yuanzhi Fu, Jinbao Zhang, Qinglian Wang

**Affiliations:** 1Henan Collaborative Innovation Center of Modern Biological Breeding, Henan Institute of Science and Technology, Xinxiang 453003, China; 2State Key Laboratory of Crop Biology, College of Agronomy, Shandong Agricultural University, Tai’an 271018, China; 3Institute of Industrial Crops, Shandong Academy of Agricultural Sciences, Jinan 250100, China

**Keywords:** *SBPase*, photosynthesis, abiotic stress, promoter, upland cotton (*Gossypium hirsutum* L.)

## Abstract

Sedoheptulose-1,7-bisphosphatase (SBPase, EC 3.1.3.37) is a key enzyme in the plant Calvin cycle and one of the main rate-limiting enzymes in the plant photosynthesis pathway. Many studies have demonstrated that the *SBPase* gene plays an important role in plant photosynthetic efficiency, yield, and stress responses; however, few studies have been conducted on the function and expression of the *GhSBPase* gene in upland cotton. In this study, our results showed that the coding sequence (CDS) of *GhSBPase* gene was 1182 bp, encoding a protein with 393 amino acids. The GhSBPase protein had adenosine monophosphate (AMP) binding site and a FIG (FBPase/IMPase/glpX) domain, and had six Cys residues and a CGGT(A/Q)C motif that were involved in redox regulation in plants. Evolutionarily, the GhSBPase protein clustered into the dicotyledon subgroup and was most closely related to the tomato SlSBPase protein. Western-blot analysis further indicated that the *GhSBPase* gene was indeed the gene encoding the SBPase protein in upland cotton. The GhSBPase protein was localized in chloroplast, which was consistent with its function as a key enzyme in photosynthesis. The *GhSBPase* gene was specifically highly expressed in leaves, and its expression level was significantly lower in a yellow-green leaf mutant than in the wild type. Moreover, the *GhSBPase* expression was in response to drought, salt, high- and low-temperature stress, and exhibits different expression patterns. The *GhSBPase* promoter had the *cis*-acting elements in response to abiotic stress, phytohormone, and light. In addition, the *GhSBPase* expression was positively correlated with the chlorophyll fluorescence parameters, suggesting that changes in the expression of the *GhSBPase* had potential applicability in breeding for enhanced cotton photosynthetic efficiency. These results will help to understand the function of the *GhSBPase* gene in photosynthesis and the adaptability of plants to external stress and provide important gene information for the high-yield breeding of crops in the future.

## 1. Introduction

Approximately 90%–95% of the dry matter comes from the organic matter synthesized through the carbon assimilation process of photosynthesis, which is the basis for the formation of plant biomass, food, fuel, and materials [1]. Improving photosynthetic efficiency to increase yield has been a major research focus in the fields of crop genetics and breeding [2,3,4]. Previously, traditional breeding has greatly improved crop yield, mainly due to improvements in agronomic measures and harvest index and rarely due to increased photosynthetic efficiency [5,6,7]. Since increasing crop yield through traditional breeding has entered a bottleneck, finding new strategies to improve crop yield has led to significant attention on photosynthesis, and breeding techniques that focus on improving photosynthetic efficiency in crops are seen as the driver for a future new “green revolution” [8,9,10].

Sedoheptulose-1,7-bisphosphatase (SBPase, EC 3.1.3.37), a key enzyme in the plant Calvin cycle and one of the main rate-limiting enzymes in the plant photosynthesis pathway, mainly acts to dephosphorylate sedoheptulose-1,7-diphosphate to regenerate ribulose 1,5-bisphosphate (RuBP), the molecular receptor for CO_2_. SBPase is located at the branch point between the regeneration stage and the assimilation stage of the Calvin cycle, that is, SBPase is related to the matter of whether carbon continues to be recycled for the regeneration of RuBP or leaves the cycle for starch biosynthesis; therefore, SBPase is very important in maintaining the flow of carbon sources during the Calvin cycle [11,12,13]. To date, the *SBPase* gene has been cloned from various species, including wheat [14], *Arabidopsis* [15], spinach [16], mulberry [17], cucumber [18], and rice [19], and its function has been extensively studied. Researchers have found that the *SBPase* gene has great potential in improving plant photosynthetic efficiency or yield. In tobacco, a slight decrease in SBPase activity can lead to a significant decrease in the carbon assimilation rate of antisense transgenic plants, and plant growth is significantly inhibited [20,21,22,23]. Similar results are also found in rice [24]. In contrast, by increasing the activity of SBPase, the photosynthetic capacity, growth rate, and biomass accumulation of transgenic tobacco and rice plants are significantly improved [25,26,27]. Transgenic wheat plants overexpressing the *SBPase* gene exhibited enhanced leaf photosynthesis, biomass, and dry seed yield [28]. In rice, overexpression of the cyanobacterial *SBPase* gene can increase the net photosynthetic rate and leaf mesophyll conductance [29,30]. Furthermore, changing the *SBPase* expression level can also improve the adaptability of plants to external stress. In rice, overexpression of the *SBPase* gene can increase the photosynthetic rate of rice seedling leaves, thus improving the tolerance to salt stress and high-temperature [31,32]. In tomato, overexpression of the *SBPase* gene can significantly increase the CO_2_ fixation rate and carbohydrate accumulation in transgenic tomato plants, thus improving the tolerance to low-temperature-induced oxidative stress [33,34]. Overexpression of the cyanobacterial *SBPase* gene in soybean can alleviate the impact of future high-temperature and high CO_2_ concentration on soybean yield [35]. In general, altering the expression and activity of SBPase may be a potential target for improving plant photosynthetic efficiency and yield in the future, especially in the face of future increasingly warming environmental conditions and increasingly higher CO_2_ concentrations, and the ability of RuBP to regenerate could be an important factor limiting photosynthetic efficiency and yield in plants [36], which makes the *SBPase* gene encoding this key enzyme limiting the regeneration rate of RuBP in the Calvin cycle even more important and becomes a new important target for improving crop yield.

Cotton is an economically important crop. As the most important breeding objective trait, high yield is closely related to the benefits of planting cotton of the cotton farmers and has always been a research area of intense focus for breeders. Photosynthetic efficiency is the main factor affecting yield and can increase cotton yield to a certain extent [37]. However, little is known about the function of the key photosynthesis gene *GhSBPase* in upland cotton. In this study, the *GhSBPase* gene in upland cotton was cloned, and the protein sequence characteristics, subcellular localization, expression characteristics, *cis*-acting elements in the promoter region, and relationships with photosynthetic traits were further investigated. These results can provide important genetic resources for improving cotton photosynthetic efficiency through genetic engineering and realizing high-yield crop breeding programs in the future.

## 2. Results

### 2.1. Identification and Cloning of GhSBPase Gene

The protein highly homologous to the *Arabidopsis thaliana* AtSBPase protein (At3g55800) in the upland cotton genome was identified by the Blastp search, and its Gene ID was Gh_D10G2468, which was named *GhSBPase* in this study. Specific primers were designed based on the coding sequence (CDS) of the *GhSBPase*, and the cDNA from the leaf of Baimian No. 1 was used as the template. A band with the same size as the CDS of the *GhSBPase* was amplified by using PCR-based methods (Figure 1A). Further sequencing showed that the PCR product was 1368 bp, of which the full-length CDS of the *GhSBPase* was 1182 bp (Figure 1B), and it was 99.92% similar to the reference genome sequence of upland cotton, indicating that the target gene was successfully cloned in our study.

### 2.2. Sequence Characteristics of GhSBPase Protein

The physicochemical properties of the GhSBPase protein were analyzed using ExPASy tools and showed that the amino acid length of the GhSBPase protein was 393 aa, and its relative molecular weight and the isoelectric point were 42.50 kDa and 5.95, respectively; the most frequently occurring amino acid was leucine (Leu) (9.4%), and tryptophan (Trp) occurred the least frequently (0.5%) (Figure 2A). The secondary structure of the GhSBPase protein was analyzed using SOPMA tools and indicated that random coils accounted for the highest proportion (43.77%), followed by alpha helix (29.52%) and extend strands (22.64%), and beta turn represented the lowest proportion (4.07%) (Figure 2B). The ChloroP 1.1 server was used to further predict the chloroplast transit peptide of the GhSBPase protein and demonstrated that the N-terminus of the GhSBPase protein had transit peptide with a length of 55 amino acids, indicating that the GhSBPase protein may function in chloroplasts (Figure 2C).

### 2.3. Conserved Domain Analysis of GhSBPase Protein

The NCBI CD-search tool was used for protein domain analysis, and the results indicated that the GhSBPase protein had the adenosine monophosphate (AMP) binding site and a FIG (FBPase/IMPase/glpX) domain, belonging to the FIG subfamily (Figure 3A). Multiple sequence alignment analysis revealed very high sequence identity between plant SBPase proteins (68.00–96.40%), indicating that these proteins were highly conserved during the evolutionary process; among them, GhSBPase had the highest identity with the tomato SlSBPase (86.80%), followed by the cucumber CsSBPase (82.70%), and the lowest identity with the maize ZmSBPase (73.50%) (Figure 3B). Moreover, the GhSBPase protein had the typical conserved features of the plant SBPase protein, such as six Cys residues and a CGGT(A/Q)C motif that were involved in redox regulation in plants (Figure 3C), suggesting that the GhSBPase protein may have function similar to those of SBPase proteins in other species.

### 2.4. Evolutionary Relationship Analysis of GhSBPase Protein

To study the upland cotton GhSBPase protein in terms of evolutionary position and relationships, a phylogenetic tree analysis on SBPase proteins from nine species was performed. The results indicated that plant SBPase proteins can be evolutionarily clustered into dicotyledon and monocotyledon subgroups (Figure 4), indicating that SBPase proteins may have existed before the evolutionary differentiation of monocotyledonous and dicotyledonous plants. Among them, the upland cotton GhSBPase protein had the closest evolutionary relationship with the SlSBPase protein in tomato, which also belonged to the dicotyledon subgroup (Figure 4).

### 2.5. Western-Blot Analysis of GhSBPase Protein

Primers containing double-digest restriction sites were designed for PCR amplification of the target gene (without the transit peptide sequence to produce the mature GhSBPase protein), and then ligated into the expression vector pCzn1. Double digestion of the recombinant expression vector pCzn1-GhSBPase contained two fragments, namely, the target gene fragment and the vector fragment, indicating that the expression vector was successfully constructed (Figure 5A). The expression strain containing the pCzn1-GhSBPase vector successfully expressed the target protein after induction with isopropyl-beta-D-thiogalactopyranoside (IPTG), and its molecular weight was close to 39.37 kDa, the predicted theoretical value of the GhSBPase protein (including His-tag) (Figure 5B). Furthermore, using an SBPase polyclonal antibody, protein extracts from cotton and *A. thaliana* leaves and *E. coli* protein extracts containing the GhSBPase recombinant protein were analyzed by western-blot. The results indicated that in both cotton and *A. thaliana* leaves, the SBPase protein band, which can react with the antibody, can be detected (positive control), while in a protein extract of *E. coli* containing the empty plasmid, the target band could not be detected (negative control); in the protein extract of *E. coli* containing the recombinant plasmid pCzn1-GhSBPase, the SBPase protein band that reacts with the antibody could be detected, with a molecular weight close to that of the SBPase protein in cotton leaves (Figure 5C). These results indicated that the *GhSBPase* gene cloned in this study was indeed the gene encoding the cotton SBPase protein.

### 2.6. Subcellular Localization of GhSBPase Protein

The WoLF PSORT tool predicted the GhSBPase protein to be localized in chloroplasts. To determine the actual subcellular localization of the GhSBPase protein, transient expression of the *GhSBPase* gene in *Arabidopsis* protoplasts was carried out. The results indicated that for the empty vector (35S::GFP), green fluorescent protein (GFP) could be detected in the plasma membrane, cytoplasm, and nucleus, while for the expression vector containing the *GhSBPase* gene (35S::GhSBPase-GFP), GFP was only detected in the chloroplast (Figure 6), indicating that GhSBPase was a chloroplast-localized protein, which was consistent with its function act as a key enzyme in photosynthesis in plants.

### 2.7. Tissue Expression Analysis GhSBPase Gene

The expression of the *GhSBPase* gene in different cotton tissues was analyzed by qRT-PCR. The results indicated that the highest expression level of *GhSBPase* was in the leaves, which were also the main location of photosynthesis, and the expression level was very low or absent in the stems, roots, petals, fibers, and boll shells (Figure 7A). We previously identified a cotton yellow-green leaf mutant with a low photosynthetic rate, and expression analysis revealed that the expression of *GhSBPase* in the mutant was significantly lower than that in the wild type (Figure 7B), indicating that the genes of photosynthetic carbon assimilation process may also be significantly affected in the yellow-green leaf mutant.

### 2.8. Analysis of the Response of the GhSBPase Gene to Abiotic Stress

The expression pattern of the *GhSBPase* gene under different abiotic stress conditions was analyzed, and the results demonstrated that the *GhSBPase* gene expression was induced in response to adverse stress, with different expression patterns under different stress conditions. Under drought stress, the *GhSBPase* gene expression was significantly up-regulated to the highest level at 3 h of treatment and then declined with prolonged treatment time (Figure 8A). Under salt stress, *GhSBPase* gene expression was significantly up-regulated to the highest level at 12 h of treatment, followed by a decrease to its lowest level at 48 h of treatment (Figure 8B). Under low-temperature stress, the *GhSBPase* gene expression continued to increase, was significantly up-regulated to its highest level at 24 h of treatment, and then declined to its lowest level at 48 h of treatment (Figure 8C). Under high-temperature stress, the *GhSBPase* gene expression showed a significant decreasing trend overall (Figure 8D).

### 2.9. Cis-Acting Elements in GhSBPase Gene Promoter

Analysis of abiotic stress-responsive *cis*-acting elements in the *GhSBPase* gene promoter showed that the promoter of this gene had a drought-responsive *cis*-acting element (MBS) (Figure 9). In addition, four phytohormone-responsive *cis*-acting elements, including one salicylic acid-responsive element (TCA-element), three abscisic acid-responsive elements (ABREs), and 21 light-responsive *cis*-acting elements, were found in the promoter of *GhSBPase* gene (Figure 9), indicating that the expression of *GhSBPase* gene might be also regulated by phytohormone and light in upland cotton.

### 2.10. Correlation between GhSBPase Gene Expression and Chlorophyll Fluorescence Parameters

The chlorophyll fluorescence parameters, such as actual photochemical quantum efficiency (ΦPSII), photochemical quenching coefficient (qP), and non-photochemical quenching coefficient (NPQ), have been widely accepted as reflecting the structure and function of the photosynthetic apparatus. To further investigate the effect of the *GhSBPase* gene expression on photosynthesis in upland cotton, we measured the *GhSBPase* gene expression level and chlorophyll fluorescence parameters in a natural population. Correlation analysis showed that the expression level of *GhSBPase* was positively correlated with chlorophyll fluorescence parameter qP (Table 1). As the chlorophyll fluorescence parameter qP reflects the level of plant photosynthetic activity, and its increase or decrease extent reflects the degree of enhancement or inhibition of photosynthetic efficiency under different conditions, the correlation between the *GhSBPase* gene expression and qP suggested that the *GhSBPase* gene could play an important role in modulating cotton photosynthetic capacity.

## 3. Discussion

Optimizing photosynthesis to improve crop yield is an effective way to solve the problem of global food security [38]. Recent studies have found that improving photosynthesis through genetic engineering can increase the yield of crops [39]. For example, overexpression of three key photosynthetic genes, violaxanthin de-epoxidase (VDE), photosystem II (PSII) subunit S (PsbS), and zeaxanthin epoxidase (ZEP) in soybean, can lead to a maximum increase of 33% seed yield, but the protein and oil contents in the seeds do not change [39]. In plants, SBPase is the key enzyme limiting the regeneration rate of RuBP during the Calvin cycle and is also one of the main rate-limiting enzymes in the plant photosynthesis pathway [40]. Many studies have demonstrated that the *SBPase* gene plays an important role in plant photosynthetic efficiency, yield, and stress responses. However, studies on the *GhSBPase* gene in upland cotton are very few. In this study, the *GhSBPase* gene was cloned by homologous cloning, and the CDS of the *GhSBPase* gene was 1182 bp, encoding a protein with 393 amino acids, and the molecular weight and isoelectric point of the GhSBPase protein were 42.50 kDa and 5.95, respectively. Previous studies have shown that plant SBPase proteins have AMP binding site and a FIG domain and have six Cys residues involved in redox regulation, and in particular, the Cys residue in the CGGT(A/Q)C motif is essential for the redox-regulated activation of SBPase proteins [33,41]. In this study, GhSBPase was found to have AMP binding site and a FIG domain, and it had six Cys residues and a CGGT(A/Q)C motif involved in redox regulation that were unique to plant SBPase proteins, indicating that the GhSBPase was consistent with the typical structural characteristics of plant SBPase proteins. Further western-blot analysis revealed that the *E. coli* protein extract containing the *GhSBPase* gene could react with an SBPase antibody, indicating that the *GhSBPase* gene was indeed the gene encoding the cotton SBPase protein. In addition, subcellular localization demonstrated that the GhSBPase protein was localized in chloroplasts; similarly, the SBPase proteins in *A. thaliana* and rice are also localized in chloroplasts [41,42]. The chloroplast localization of the GhSBPase protein observed in this study was consistent with its function as a key enzyme responsible for the regeneration rate of RuBP in the chloroplast during the Calvin cycle. Previous studies have found that most chloroplast-localized proteins have a transit peptide at the N-terminus that enables the protein to localize to the chloroplast [41], and the transit peptide not only transports chloroplast proteins but also transports target proteins of interest into the chloroplast. To date, improvements in the expression of foreign proteins by chloroplast transit peptides have been reported in the small subunit of ribulose 1,5-bisphosphate carboxylase/oxygenase (Rubisco) [43,44]. This study identified a transit peptide sequence at the N-terminus of the GhSBPase protein, which may be crucial for the GhSBPase protein to enter the chloroplast to function. In future studies, if the transit peptide sequence of the GhSBPase protein can be cloned and linked to a metabolite gene of interest and the chloroplast can be used as an organelle for mass production, it may be of great significance for increasing metabolite content.

In plants, the expression of the *SBPase* gene is mainly regulated by light, developmental stage, and glucose regulation [15,45,46] and is tissue-specific [18]. This study found that the *GhSBPase* gene was specifically highly expressed in leaves and expressed at low levels or was absent in other tissues, indicating that the *GhSBPase* gene may play an important role in photosynthesis in cotton leaves. Similarly, the rice *SBPase* gene also shows a leaf-specific expression pattern [41]. Previous studies have shown that the promoters of genes specifically expressed in leaf tissues can be used to achieve the directional expression of target genes in leaf tissues [47]. For example, the promoters of the key enzymes SBPase and fructose-1,6-bisphosphatase aldolase (FBPA) in the Calvin cycle can be used to achieve the specific expression of target genes in leaf mesophyll cells [47]. This study found that *GhSBPase* was a gene specifically expressed in leaf tissues, indicating that the promoter of this gene may also be important in achieving the directional expression of target genes in leaf tissues. In addition, this study found that *GhSBPase* gene expression was significantly down-regulated in the leaves of a yellow-green leaf mutant, indicating that in the mutant plant, there might be an adverse effect on the photosynthetic carbon assimilation process and the expression of photosynthetic genes in cotton plants due to the blocked chlorophyll synthesis. Interestingly, the correlation between *GhSBPase* gene expression and the chlorophyll fluorescence parameter qP was observed in our study, which suggested that the *GhSBPase* gene could play an important role in modulating cotton photosynthetic capacity.

Light is the energy source for plants to perform photosynthesis. Studies have found that changing light intensity, quality, and duration can accelerate photosynthesis and flowering, which can shorten breeding years and realize accelerated breeding [48]. This study found that the promoter region of the *GhSBPase* gene had many light-responsive *cis*-acting elements, indicating that the expression of this gene may be regulated by light. Similarly, in rice, low-light stress can induce significantly up-regulated *SBPase* gene expression [49]; in contrast, dark treatment can significantly down-regulate *SlSBPase* expression in tomato and severely inhibit the activity of SBPase [50]. Furthermore, in wheat, dark treatment can cause sharply decreased *SBPase* promoter activity, but *SBPase* promoter activity can return to normal levels when the plants are under light for 3 h; interestingly, the light intensity does not affect the activity of the *SBPase* promoter [51]. In addition, some studies have demonstrated that phytohormone treatment can improve photosynthesis and adaptability to stress in plants. For example, gibberellin treatment can improve the photosynthesis and yield of wheat [52], salicylic acid treatment can enhance the stability of the photosystem II (PSII) in wheat leaves under cold stress conditions [53], and abscisic acid treatment can slow the decline in the photosynthetic rate under salt stress conditions and enhance the tolerance of broad bean to salt stress [54]. This study found that there was an abscisic acid-responsive *cis*-acting element and a salicylic acid-responsive *cis*-acting element in the *GhSBPase* promoter, indicating that the expression of this gene may also be regulated by phytohormone. To date, there have been reports on *SBPase* gene expression regulation by phytohormone. For example, in tomato, the phytohormone methyl jasmonate (MeJA) can significantly inhibit the expression of the tomato *SlSBPase* and induce the expression of senescence-related genes [50]. Currently, it is not clear how phytohormone affects *GhSBPase* gene expression in cotton, which still must be further studied in the future.

A key factor limiting crop yield is abiotic stress on plants, which severely affects the photosynthetic capacity of crops, and plants have evolved complex systems to cope with various environmental challenges [55]. Previous studies have found that regulation of the expression of photosynthetic genes plays an important role in the response of plants to abiotic stress [33,34,56]. In this study, we found that the expression of *GhSBPase* in upland cotton was affected to varying degrees under drought, salt, and low- and high-temperature stress conditions. Under drought stress conditions, the *GhSBPase* gene expression was significantly up-regulated, so *GhSBPase* may play an important role in the response of cotton plants to drought stress. Consistent with this potential role, the *GhSBPase* gene promoter region contained a drought-responsive *cis*-acting element (MBS). In addition, this study found that salt stress also induced a significant up-regulation of *GhSBPase* expression, suggesting that *GhSBPase* may also play an important role in the cotton plants response to salt stress. Similar results have also been observed in *Suaeda liaotungensis* [57]. In addition, in rice, the accumulation of SBPase in vivo is also considered to have a new physiological function, that is, to protect photosynthesis from salt stress, and overexpression of the *SBPase* gene is beneficial for improving the photosynthesis and growth of transgenic plants under salt stress [31]. Compared with other abiotic stresses, temperature stress has a significantly adverse effect on crop yield [58]. Our study found that *GhSBPase* expression was up-regulated under low-temperature stress, indicating that *GhSBPase* may play an important role in maintaining the photosynthetic efficiency under low-temperature stress conditions. Likewise, overexpression of the *SlSBPase* gene can significantly increase photosynthetic efficiency and tolerance to low-temperature stress in transgenic tomato plants [33]. In addition, a melatonin-mediated increase in SBPase activity can improve tomato photosynthetic carbon fixation capacity, which can effectively alleviate the impact of low-temperature on tomato production [59]. However, interestingly, under high-temperature stress, *GhSBPase* showed a different expression pattern from that under other stress conditions. Under high-temperature stress, *GhSBPase* expression was continuously down-regulated, indicating that high-temperature stress may have adverse effects on the carbon assimilation process during cotton photosynthesis. Previous studies have reported that moderate heat stress inhibits leaf photosynthesis, resulting in a decrease in the net photosynthesis (Pn) and a decrease in Rubisco activity [60]. When the temperature exceeds 35 °C, the Calvin cycle cannot function properly [61]; transcriptional repression of photosynthetic genes is also considered a general response to heat stress [62,63]. However, some studies have found that post-transcriptional regulation may also play an important role in the adaptation of photosynthetic genes to high-temperature stress. For example, in wheat, heat stress-induced changes in the activity of the photosynthetic gene Rubisco activase (RCA) are largely regulated at the post-transcriptional level [63]. It is unclear whether post-transcriptional regulation is involved in the change in SBPase activity in cotton leaves to enhance the adaptability of cotton plants to high-temperature stress. In general, this study found that *GhSBPase* gene expression is greatly affected by environmental factors, which is consistent with the findings of previous studies that the plant *SBPase* gene is involved in plant responses to various stresses, and the application of the *SBPase* gene in breeding may be important for improving yield and stress tolerance in cotton.

## 4. Materials and Methods

### 4.1. Plant Materials, Growth Conditions and Stress Treatment

The plant materials used in this study were the upland cotton variety Baimian No.1. For the tissue expression analysis, the leaves, stems, roots, petals, boll shells, and 20 DPA (Day post anthesis, DPA) fibers of cotton plants were collected under field conditions. For the stress expression analysis, the cotton seeds were sown in nutrient soil and cultured in a growth chamber. When the cotton seedlings grew to one true leaf stage, they were transferred to 1/2 Hoagland’s nutrient solution for 7 days. Then, the cotton seedlings with consistent growth were selected for drought (20% PEG6000), salt (200 mM NaCl), low-temperature (4 °C) and high-temperature (38 °C) stress treatment. After 0 h (CK), 3 h, 6 h, 12 h, 24 h, and 48 h of treatment, the leaves of cotton seedlings were sampled and frozen in liquid nitrogen and were stored at −80 °C for RNA extraction.

### 4.2. RNA Extraction and cDNA Synthesis

Total RNA was extracted using the plant RNA extraction kit (DP441, TIANGEN, Beijing, China). First strand cDNA was synthesized from RNA according to the manual of reverse transcription kit (6210A, TaKaRa, Dalian, China).

### 4.3. Isolation of GhSBPase Gene from Cotton Leaf

The specific primers sequences were designed by using the Primer 5.0 software based on the sequence information of *GhSBPase* gene, as follows: GhSBPase-F (5’−3’): CTCAAAGATTCATATCAAAAGT; GhSBPase-R (5’−3’): AAATTTATATGAAAAGTACTTAAAG. Using the cDNA of cotton leaf as the template, the PCR amplification was performed with PrimeSTAR GXL DNA Polymerase (R050A, TaKaRa, Dalian, China), and the PCR reaction system was carried out according to the instructions. The PCR amplification procedure included pre-denaturation at 94.0 °C for 2 min, followed by 33 cycles, 98.0 °C for 20 s and 53.0 °C for 1 min 30 s, and a final extension cycle of 72 °C for 10 min. The PCR product was separated by gel electrophoresis and purified using the DNA purification kit (DP219, TIANGEN, Beijing, China). Then, a poly A-tail was added to the PCR product, and the product was ligated into the pMD19-T vector to transform *DH5α* competent cells and verified by sequencing.

### 4.4. Sequence Analysis of GhSBPase Protein

The ExPASy (https://web.expasy.org/protparam/, accessed on 8 September 2017) was used for prediction analysis of the molecular weight, isoelectric point, amino acid length, and composition of protein. The NCBI CD-search (https://www.ncbi.nlm.nih.gov/Structure/cdd/wrpsb.cgi/, accessed on 16 September 2017) was used to predict the conserved domains of protein. The SOPMA (https://npsa-prabi.ibcp.fr/cgi-bin/npsa_automat.pl?page=n psa_sopma.html/, accessed on 16 September 2017) was applied for the prediction of protein secondary structures. The protein chloroplast transit peptide analysis was performed by using ChloroP 1.1 server [64]. The WoLF PSORT (https://www.genscript.com/wolf-psort.html?src=leftbar/, accessed on 17 September 2017) was used for protein subcellular localization prediction. The sequence identity of protein was completed by BioEdit software [65]. Multi-sequence alignment of proteins was completed by Clustal X2 software [66]. A phylogenetic tree was constructed by the Neighbor-Joining (NJ) method with 1000 bootstrap replicates in MEGA 5.02 software [67], the *p*-distance model was selected, and pairwise deletion was used to deal with missing data.

### 4.5. SDS-PAGE and Immunological Analysis

The *GhSBPase* gene (excluding chloroplast transit peptide) was cloned between the *Sal* I and *Xba* I sites of expression vector pCzn1. Then, the recombinant expression vector pCzn1-GhSBPase was transformed to *Escherichia coli* Arctic Express, and GhSBPase protein was induced expression by 0.5 mM isopropyl-beta-D-thiogalactopyranoside (IPTG) for 4 h at 37 °C. The *E. coli* proteins were separated by 12% SDS-PAGE and transferred to Coomassie Brilliant Blue staining solution for dyeing. For western-blot analysis, the total protein from *Arabidopsis thaliana* and cotton leaf, *E. coli* protein, *E. coli* proteins containing GhSBPase recombinant protein were separated by 12% SDS-PAGE and then transferred onto a PVDF membrane. The membrane was incubated overnight at 4 °C with polyclonal SBPase antibodies (AS152873, Agrisera, Vännäs, Sweden) at a dilution of 1:10,000 and incubated 1 h at 37 °C with the second antibody at a dilution of 1:5000. Images of the blots were obtained using a CCD camera imaging system.

### 4.6. Subcellular Localization

The pAN580-GFP vector was double digested using the restriction enzymes Xba I and BamH 1. The specific primers were designed to amplify the GhSBPase gene, and the specific primers were as follows: GhSBPase-GFP-F (5’−3’): AAGTCCGGAGCTAGCTCTAGAATGGAGACTAGTGTCACGTG; GhSBPase-GFP-R (5’-3’): GCCCTTGCTCACCATGGATCCAGCAGTAGCTCCAACAGGG (The underline indicates homologous arm). The pAN580-GhSBPase-GFP vector was constructed by homologous recombination. Then, the constructed expression vector pAN580-GhSBPase-GFP and the empty vector pAN580-GFP were transformed into *Arabidopsis* protoplasts, respectively. The transformed protoplasts were cultured under dark conditions for 12 h to 16 h, and then the green fluorescent protein (GFP) signal in the protoplasts was observed by using the confocal scanning microscope (LSM780, Zeiss, Oberkochen, Germany).

### 4.7. Quantitative Real-Time PCR (qRT-PCR) Analysis

qRT-PCR analysis was performed by the SYBR Green dye method with the *Actin* gene as an internal reference to determine gene relative expression levels using the 2^−ΔΔCT^ method [68]. The specific primers used for *GhSBPase* gene expression analysis were designed by using the Primer 5.0 software, and the specific primers were as follows: GhSBPase-Q-F (5’-3’): TGTTCTTCCTGGTGTCTC; GhSBPase-Q-R (5’-3’): CTAATCAATGCCTTATCTGT; Actin-Q-F (5’-3’): GACCGCATGAGCAAGGAGAT; Actin-Q-R (5’-3’): GCTGGAAGGTGCTGAGTGAT. The amplification program was 95 °C for 30 s for 1 cycle and 95 °C for 5 s and 60 °C for 20 s for 40 cycles. Three independent replicates were set for each sample and amplified on a fluorescence quantitative PCR instrument (FQD-48A, BIOER, Hangzhou, China).

### 4.8. Cis-Acting Elements Analysis in the Promoter Region

The *GhSBPase* gene promoter sequence (2000 bp upstream of the ATG) was obtained from the CottonFGD (https://cottonfgd.org/, accessed on 10 June 2022). Using the PlantCare tools (http://bioinformatics.psb.ugent.be/webtools/plantcare/html/, accessed on 12 June 2022), the *cis*-acting elements in the promoter region were analyzed.

### 4.9. Trait Measurement and Statistical Analysis

The chlorophyll fluorescence parameters such as Φ_PSII_, qP, and NPQ (see Table 1 for definitions), which reflect photosynthetic system activities, and the expression level of the *GhSBPase* gene, were measured using the upper-third leaf of the cotton plants at the flowing stage in a natural population of upland cotton [69]. The chlorophyll fluorescence parameters were measured by a PAM fluorometer (DUAL-PAM-100, Heinz Walz, Effeltrich, Germany) using a previously described procedure [70]. The Pearson phenotypic correlations among the traits were calculated using SPSS 17.0 software.

## Figures and Tables

**Figure 1 ijms-24-06648-f001:**
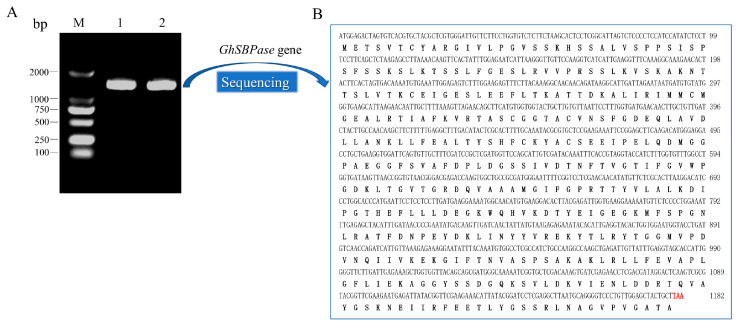
Cloning of *GhSBPase* gene in upland cotton. (**A**) Agarose gel electrophoretic analysis of PCR product of *GhSBPase* gene. M: DNA marker (DL2000); 1 and 2: PCR product of *GhSBPase* gene. (**B**) The *GhSBPase* gene CDS sequence and its encoded protein by sequencing. Red base represents the stop codon.

**Figure 2 ijms-24-06648-f002:**
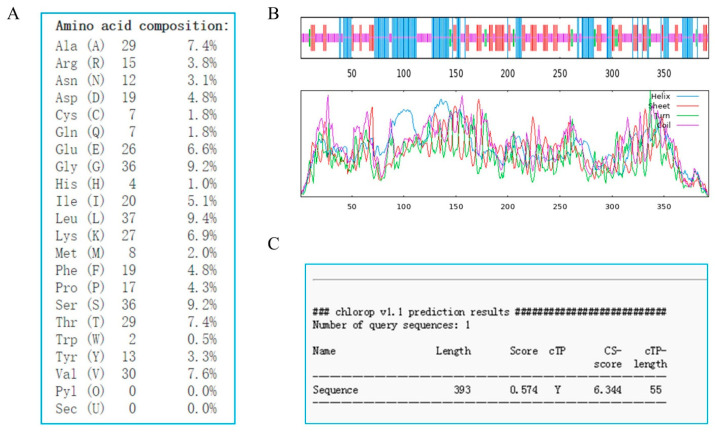
Sequence characteristics analysis of GhSBPase protein in upland cotton. (**A**) Analysis of the amino acid composition of GhSBPase protein. (**B**) Analysis of the secondary structure of GhSBPase protein. The blue, green, purple and red lines represent alpha helix, beta turn, random coil and extended strand respectively. (**C**) Prediction analysis of chloroplast transport peptide of GhSBPase protein.

**Figure 3 ijms-24-06648-f003:**
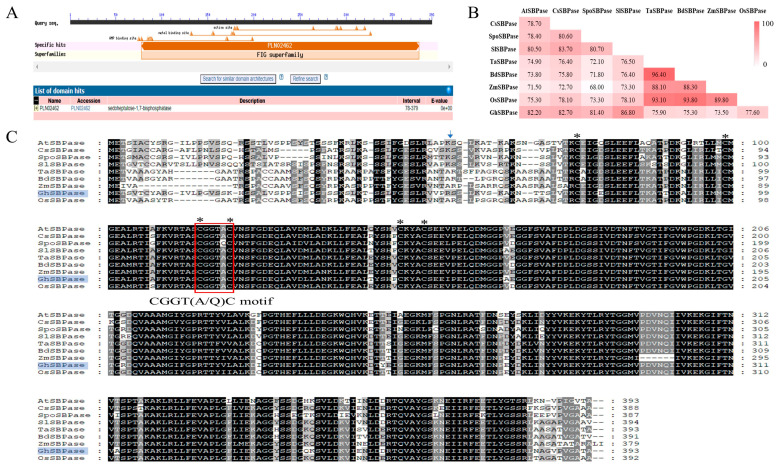
Conserved domain and sequence identity analysis of GhSBPase protein in upland cotton. (**A**) Conserved domain prediction of GhSBPase protein. (**B**) The sequence identity between GhSBPase protein and SBPase proteins in other plants. (**C**) Muti-sequence alignment of plant SBPase proteins. The symbol * represents cysteine; the blue arrow refers to the cutting site of the chloroplast transport peptide of GhSBPase protein; the red box indicates CGGT(A/Q)C motif.

**Figure 4 ijms-24-06648-f004:**
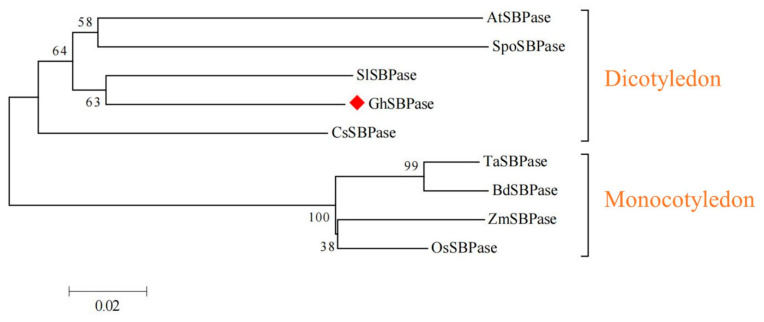
Phylogenetic tree analysis of SBPase protein from upland cotton and other plants. The numbers on the branches represent the reliability percent of bootstrap values based on 1000 replication; the symbol ◆ represents GhSBPase in upland cotton; *Arabidopsis thaliana*, AtSBPase (AEE79443); *Oryza sativa*, OsSBPase (AAO22558); *Zea mays*, ZmSBPase (ONM36378.1); *Triticum aestivum*, TaSBPase (CAA46507); *Solanum lycopersicum*, SlSBPase (FJ959073); *Cucumis sativus*, CsSBPase (ACQ82818); *Spinacia oleracea*, SpoSBPase (O20252.1), and *Brachypodium distachyon*, BdSBPase (XP_003564625.1).

**Figure 5 ijms-24-06648-f005:**
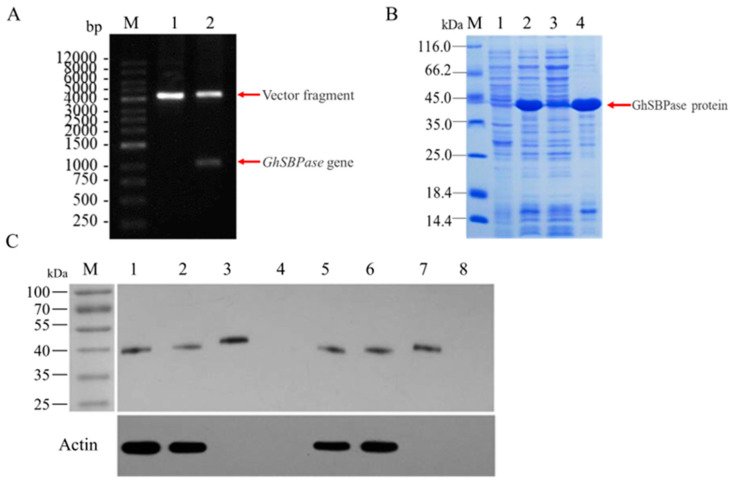
Western-blot analysis of GhSBPase protein in upland cotton. (**A**) Restriction enzyme digestion of expression vector pCzn1-GhSBPase. 1: Plasmid of expression vector before enzyme digestion; 2: Plasmid of expression vector after enzyme digestion. (**B**) SDS-PAGE analysis of GhSBPase recombinant protein. 1: Uninduced *Escherichia coli* protein extracts; 2: *Escherichia coli* protein extracts after IPTG induction; 3: Cleaning of *Escherichia coli* protein extracts after IPTG induction; 4: Precipitation of *Escherichia coli* protein extracts after IPTG induction. (**C**) Western-blot detection of GhSBPase proteins. 1 and 5: *Arabidopsis* leaf protein; 2 and 6: Cotton leaf protein; 3 and 7: GhSBPase recombinant protein; 4 and 8: Empty bacterial protein.

**Figure 6 ijms-24-06648-f006:**
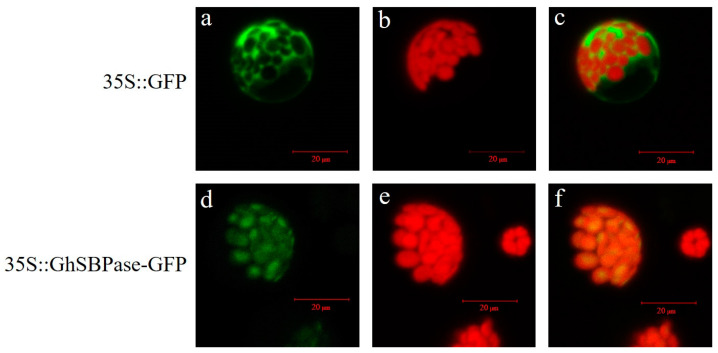
Subcellular localization of GhSBPase protein in upland cotton. Confocal microscope images of the GFP fluorescence (**a**,**d**), chlorophyll auto fluorescence (**b**,**e**), and the merged images for GFP fluorescence and chlorophyll auto fluorescence (**c**,**f**).

**Figure 7 ijms-24-06648-f007:**
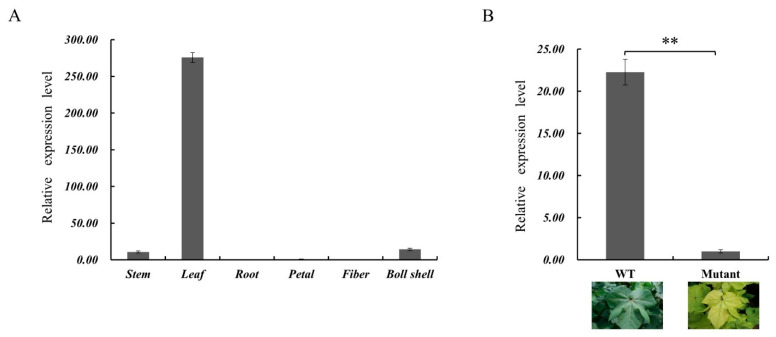
Expression analysis of *GhSBPase* gene in different tissues (**A**) and yellow-green leaf mutant (**B**) of cotton. ** *p* < 0.01.

**Figure 8 ijms-24-06648-f008:**
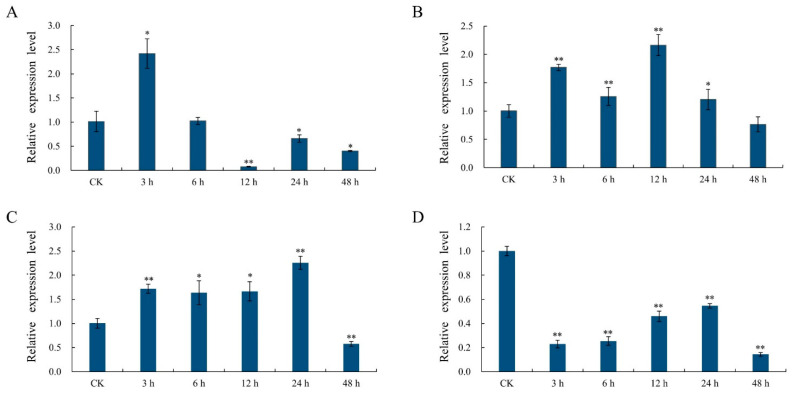
Stress expression analysis of *GhSBPase* gene in upland cotton. (**A**) drought treatment; (**B**) salt treatment; (**C**) low-temperature treatment; (**D**) high-temperature treatment. * *p* < 0.05; ** *p* < 0.01.

**Figure 9 ijms-24-06648-f009:**
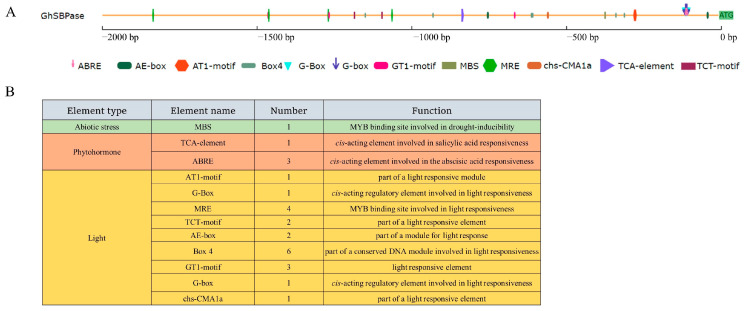
*Cis*-acting elements analysis of *GhSBPase* promoter in upland cotton. (**A**) The position distribution diagram of *cis*-acting elements in *GhSBPase* promoter, and different colors represent different kinds of *cis*-acting elements. (**B**) Category and number of *cis*-acting elements in *GhSBPase* promoter.

**Table 1 ijms-24-06648-t001:** Correlation coefficients among *GhSBPase* expression and chlorophyll fluorescence parameters in a natural population.

Traits	Φ_PSII_	qP	NPQ
*GhSBPase* expression	0.076	0.248 **	−0.126

Φ_PSII_: actual photochemical quantum efficiency; qP: photochemical quenching coefficient; NPQ: non-photochemical quenching coefficient; ** *p* < 0.01.

## Data Availability

The data presented in this study are available on request from the corresponding author.

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
