# Peer review of "Sequence Characteristics and Expression Analysis of the Gene Encoding Sedoheptulose-1,7-Bisphosphatase, an Important Calvin Cycle Enzyme in Upland Cotton (Gossypium hirsutum L.)"

_ijms, 2023, doi:10.3390/ijms24076648_

Round 1

Reviewer 1 Report

The use of the phrase 'expression patern' as a key word seems to me inappropriate. This is not a phrase that distinguishes this article/topic in any way. It seems to be characteristic of many scientific publications (in this one, it is not even the main topic of achievement). Phrased so unspecifically, it only increases the likelihood that the article will come up in a search on the Internet.

Author Response

#Reviewer 1

Comment 1: The use of the phrase 'expression patern' as a key word seems to me inappropriate. This is not a phrase that distinguishes this article/topic in any way. It seems to be characteristic of many scientific publications (in this one, it is not even the main topic of achievement). Phrased so unspecifically, it only increases the likelihood that the article will come up in a search on the Internet.

Answer: Many thanks for reviewer’s helpful suggestions. In our revised manuscript, we have changed the key word “expression pattern” to “promoter” (line 36).

Reviewer 2 Report

The presented article is devoted to the complex characterization of cotton Sedoheptulose-1,7-bisphosphatase encoding gene. Its structure, expression and similarities with ortolog genes in other plants were analyzed using different molecular-engineering methods. It was discovered that the GhSBPase protein is localized in chloroplast, and functions as a key enzyme in photosynthesis. Evidently the GhSBPase gene was specifically highly expressed in green leaves of wild type plants, and its expression level was significantly lower in a yellow-green leaf mutant. Also, the GhSBPase gene expression was involved in plant responses to drought, salt, high or low temperatures stress. It was discovered that GhSBPase promoter had the cis-acting elements in response to abiotic stress, phytohormone, and light. It was shown that GhSBPase expression positively correlates with chlorophyll fluorescence parameters, which propose that changes in the expression of GhSBPase have potential applicability in breeding for improved cotton photosynthetic productivity. These results have high theoretical and practical value.

At the same time I have some remarks.

1. Page 3, line 114; page 4, line 166: from where did you take the primers sequences?

2. Not all abbreviations are explained in the text, or may be, it is difficult to find their decoding. May be it is better to insert the list of abbreviations.

Author Response

#Reviewer 2

The presented article is devoted to the complex characterization of cotton Sedoheptulose-1,7-bisphosphatase encoding gene. Its structure, expression and similarities with ortolog genes in other plants were analyzed using different molecular-engineering methods. It was discovered that the GhSBPase protein is localized in chloroplast, and functions as a key enzyme in photosynthesis. Evidently the GhSBPase gene was specifically highly expressed in green leaves of wild type plants, and its expression level was significantly lower in a yellow-green leaf mutant. Also, the GhSBPase gene expression was involved in plant responses to drought, salt, high or low temperatures stress. It was discovered that GhSBPase promoter had the cis-acting elements in response to abiotic stress, phytohormone, and light. It was shown that GhSBPase expression positively correlates with chlorophyll fluorescence parameters, which propose that changes in the expression of GhSBPase have potential applicability in breeding for improved cotton photosynthetic productivity. These results have high theoretical and practical value.

At the same time, I have some remarks.

Comment 1: Page 3, line 114; page 4, line 166: from where did you take the primers sequences?

Answer: The specific primers sequences were designed by using the Primer5.0 software based on the sequence information of GhSBPase gene. In the original manuscript, the description of primers sequences was not clear enough, and it has been modified in our revised manuscript (lines 113-118, lines 167-168).

Comment 2: Not all abbreviations are explained in the text, or may be, it is difficult to find their decoding. Maybe it is better to insert the list of abbreviations.

Answer: Based on helpful comments from the reviewer, we have checked all abbreviations in the text, and supplemented the explanations for all abbreviations in our revised manuscript (lines 18-19, lines 20, line 143, line 162, lines 360-361, line 390, line 404, line 471, line 477).